# Comparative performance of public and private primary care service delivery in Malaysia: An analysis of findings from QUALICOPC

Su Miin Ong[1‡]*, Ming Tsuey Lim[1‡], Seng Fah Tong[2], M. N. Kamaliah[3], Peter Groenewegen[4,5], Sheamini Sivasampu[1]

1 Centre for Clinical Outcomes Research, Institute for Clinical Research, Shah Alam, Selangor, Malaysia, 2 Department of Family Medicine, Faculty of Medicine, Universiti Kebangsaan Malaysia, Hospital Canselor Tuanku Muhriz UKM, Wilayah Persekutuan Kuala Lumpur, Malaysia, 3 Faculty of Medicine, University of Cyberjaya, Cyberjaya, Selangor, Malaysia, 4 Netherlands Institute for Health Services Research (NIVEL), Utrecht, The Netherlands, 5 Departments of Human Geography and Department of Sociology, Utrecht University, Utrecht, The Netherlands

‡ SMO and MTL should be considered joint first authors.
* ongsm@crc.gov.my

**Data Availability Statement:** Data cannot be shared publicly because of the study data contain potentially sensitive information and the restriction

## Abstract

### Introduction

Primary care services are essential in achieving universal health coverage and Malaysia is looking into public-private partnership to overcome resource constraints. The study aims to compare the performance of primary care service delivery dimensions between public and private sector.

### Methods

This cross-sectional study used the data from the Malaysian International Quality and Costs of Primary Care (QUALICOPC) study conducted in 2015–2016. The relative performance of each sector in four dimensions was compared using multi-level linear regression by incorporating a dummy variable indicating public sector in the model.

### Results

The public sector was shown to have higher performance in comprehensiveness and coordination, while the private sector was better in continuity. There was no significant difference in accessibility. The public primary care services were better in serving primary care sensitive conditions, better informational continuity, and with better skill-mix and inter- and intra-professional relationship. Meanwhile, the private sector was stronger in referral decision making process, specialist feedback and greater out of hours facilities access.

under the Personal Data Protection Act 2010 Malaysia. Data are available from the Institute for Clinical Research (contact via contact@crc.gov.my or sheamini@crc.gov.my) for researchers who meet the criteria for access to confidential data.

**Funding:** The study was supported by the grant from Ministry of Health Malaysia under the Malaysian Health System Research Initiative (MHSR). This study is a sub-work package under the MHSR initiative registered under the National Medical Research Registry with approval number NMRR 15-607-25769. The funders had no role in study design, data collection and analysis, decision to publish, or preparation of the manuscript. OSM, LMT and SS are employees of Ministry of Health Malaysia at the time of manuscript preparation.

**Competing interests:** The authors have declared that no competing interests exist.

## Conclusions

The public and private sectors differ in their strengths, which the government may tap into to strengthen primary care services. Other areas for improvement include seamless care strategies that promote good referral, feedback, and information continuity.

## Introduction

Strong primary care is essential to achieving universal health coverage, a principal goal for the global health community through 2030 [1]. Primary care is generally a patient's first point of contact with the health care system. It can be provided through public or private providers. Private primary care provision is expanding in both low- and middle-income countries and high-income countries. [2–4]. The growth is mainly due to challenges faced, such as financial constraints, shifts in disease burden from communicable to chronic non-communicable diseases, demographic shift, population displacement, and political and economic instability [5]. This has resulted in many countries having mixed health systems, that comprised of public and private providers, to compliment the constraints faced by public sector [6]. At the same time, the World Bank has recommended a policy of reducing government involvement in healthcare and promotion of private sector in the provision of healthcare services [7, 8]. The trend towards higher levels of participation of the private sector is to improve the efficiency of service provision including meeting the growing healthcare needs [7, 9].

Primary care services in Malaysia exist in two parallel system where the public primary care clinics are governed mainly by the Ministry of Health of Malaysia (MOH), while the private primary care clinics are privately owned practices. The public sector is financed mainly through general revenue and taxation collected by the government, while the private sector is funded principally through out-of-pocket payments from patients, employers as part of employee health benefits, and some private health insurance [10]. Even though the number of private practices outweigh public clinics in a ratio of six to one [11], higher patient visits were recorded in public clinics [12]. The private practices' workforce mainly consist of doctors and non-certified nursing aides [12]. Three quarters of these clinics are solo practices too [11]. In comparison, most public clinics are group practices with a skill-mix of primary care providers such as family medicine specialists, doctors, physiotherapists, occupational therapists, certified nurses, and pharmacists [12]. In addition, the public clinics in Malaysia have a wide geographical coverage and offer a comprehensive range of services, including health promotion, disease prevention, curative and rehabilitative care. The private health sector provides health services, mainly in urban, affluent areas with a focus on curative care [10]. These private clinics are subjected to the Private Healthcare Services and Facilities Act and Regulations, which is largely related with qualifications and structural aspects of operation [10]. However, the enforcement of the Act is unsatisfactory [10]. Furthermore, there are fewer processes to monitor in the private sector. In contrast, the public clinics are subject to standard operating procedures and the quality of these clinics are monitored by MOH using a national set of key performance indicators [10].

In countries with mixed health systems, the private sector is often perceived to offer access to greater service capacity and responsiveness, managerial expertise, technology and innovation as well as investment and funding [6]. Hence, it is imperative to understand how the quality and performance in the private sector compares with that in the public sector. It is important to have an optimal balance of roles and facilities in the provision of health services

between public and private sectors. By identifying the strengths and weaknesses of the service delivery process of the two sectors, the gaps between the sectors can be bridged in anticipation of possible future public-private partnership and lead towards a better holistic health system for all patients.

The differences in the provision of primary care between the public and private sectors in terms of organisation, source of finances, governance and regulations, and in provider and patient characteristics [13, 14] may affect the quality of primary care service delivery in countries with mixed health system. The participation of Malaysia in the International Quality and Costs of Primary Care (QUALICOPC) study which reflects on the current state of primary care in Malaysia, allows comparison between the public and private sectors to be made. This is a rare instance where both sectors were examined using a standardised questionnaire, with the same method and study period. Therefore, in this study, we aimed to use the data from QUALICOPC to answer the question, is there a difference in primary care services delivery dimensions between public and private clinics from the perspective of providers? Specifically, in the dimensions of accessibility, comprehensiveness, continuity and coordination of care.

## Materials and methods

### Data source

The data used in this study were from the QUALICOPC study conducted in Malaysia. This study was approved by the Medical Research and Ethics Committee (MREC), Ministry of Health Malaysia. All participants gave informed consent to be interviewed. QUALICOPC is an international comparative survey that was conducted in 31 European countries, Australia, New Zealand, Canada, and in Malaysia. It consists of four sets of questionnaires that gather information about the primary care settings, practitioners, services and patients. The questionnaires were validated by family medicine specialists and healthcare researchers for face and content validity. We have used the same measurements in our previous study which demonstrated the differences between countries for these concepts, thus providing evidence for its content validated [15]. For this study, we used the data from the practitioner's questionnaires, which collected information on the primary care settings and service provision.

QUALICOPC in Malaysia was conducted in two phases; 2015 in the public sector, and 2016 in the private sector. The list of primary care clinics was obtained from the Ministry of Health Malaysia through the Family Health Development Division for public clinics and the Private Medical Practice Control Division for private clinics. If a sampled clinic had more than one doctor working on that day of data collection, only one primary care doctor per clinic would be selected to participate in the study. The doctor in the public clinics was selected by simple random sampling, whereas in the private clinics, the doctor was selected by convenient sampling [16]. A total of 222 public primary care practitioners were sampled with 221 responded (response rate of 99.5%). As for private sector, 239 practitioners responded to the survey out of the 510 clinics sampled, giving a response rate of 46.9%. The calculation was made based on the target sample size of 220 clinics similar to other countries which also participated in QUALICOPC survey for benchmarking. We are also taking into account the high non-response rate of a previous local study among public and private clinics in Malaysia [11]. Hence, the sample size for private clinics was inflated from targeted 220 to overcome the low response rate in private clinics. Various strategies including obtaining support letters from the Deputy Director General of Health and Technical Support (DDG), state health directors as well as the Malaysian Medical Association (MMA) and the Board of Directors of large chain clinics were taken to increase the doctor's participation in the private sector. The detailed conduct of the study were published elsewhere [17, 18].

## Questionnaires

Information about different primary care service delivery dimensions were collected in the practitioner's questionnaire [19]. For this study, we identified 33 indicators from the practitioner's questionnaire, and mapped it to the four dimensions of accessibility, comprehensiveness, continuity, and coordination as defined by World Health Organization (WHO) [20]. These indicators were selected and mapped based on the dimensions classified during the questionnaire development of QUALICOPC [19]. Table 1 displays the details of the indicators together with their corresponding dimensions and the scale they were measured in. The questionnaire was included as a S1 File.

## Analysis

A descriptive analysis was performed to compare the demographics characteristics between the public and private primary care practitioners. Chi-squared tests were done for categorical variables while age was compared using Kruskal-Wallis test.

As the indicators were measured on different scales, the responses were first converted to a common scale between 0 and 1 using min-max normalisation method. This method of normalisation will preserve the relationship among the original data [21]. After the data pre-processing stage, multiple items responses for each indicator were combined using ecometric approach via multilevel linear regression models [22]. Item responses will be nested within individual practitioner. A separate model will also be computed for each dimension using the same principle, with an additional level of indicators, in each item responses are nested within dimensions and within individual practitioners as the highest level.

For comparison between sectors, a dummy variable indicating public clinics was added to each model. This will allow us to directly compare the relative performance between public and private clinics through the coefficient of this dummy variable (public sector coefficient, $\beta_{public}$). Sensitivity analysis that included other independent variables such as age and sex of the practitioners, and practice locations was also conducted to examine the robustness of the models. All analysis was performed using R version 3.6 [23] and the multilevel analysis was done using the lme4 [24] package in R.

# Results

## Demographics

The demographics of the practitioners participating in the study are shown in Table 2. The practitioners in the private sector were significantly older than those in the public sector. While the private sector practitioners were predominantly men, there were more women in the public sector. More than half of the private clinics were located in city or suburb, and only about a fifth of the public clinics were located in those area.

## Accessibility

There was a total of eight indicators to measure accessibility in QUALICOPC (Fig 1). Among them, four indicators were found to be not significantly different between the two sectors. Private sector clinics were in closer proximity to other healthcare centres, offering better extended after-hours visits, and seeing fewer financial defaulters. However, the public sector was rated higher in terms of providing healthcare services to the uninsured. Overall, for the whole dimension, there was no significant difference observed between the two sectors in accessibility ($\beta_{public}$ = -0.010, p = 0.344).

**Table 1. List of indicators and its respective dimensions.**

| Dimension[†] | Indicators[‡] | Code | Scale |
|---|---|---|---|
| Accessibility | Physical distance between healthcare facilities | Distance | 1–4 |
| Definition: Available to all with minimum delay and not limited by geographical, cultural, administrative or financial barriers. General population regardless of age, gender, race, religion, socioeconomic status or any health-related problem can readily access the services. | Availability of after hour visit to primary care practice | After hours | 0–1 |
| | After hour medical services availability | Service after hours | 1–5 |
| | Weekend medical services availability | Service weekend | 1–5 |
| | Provides walk-in hours for patients | Walk-in | 0–1 |
| | Support for patients with financial obstacles | Financial aid | 0–1 |
| | Frequency of defaulters due to financial constraints | Financial default | 1–3 |
| | Restriction in accepting new patients into practice | Restriction | 0–1 |
| | Providing services without remuneration | Uninsured | 1–4 |
| Comprehensiveness | Availability of medical equipment | Medical equipment | 0–1 |
| Definition: Provides integrated health services that include health promotion, disease prevention, curative, rehabilitative and supportive care to patients. | Access to laboratory facilities | Lab access | 1–3 |
| | Access to X-ray facilities | X-ray access | 1–3 |
| | First contact for common ailments | First contact | 1–4 |
| | Treatment and follow-up with diagnosed patients | Treatment | 1–4 |
| | Minor medical procedures in practice | Procedures | 1–4 |
| | Measuring of blood pressure in patients | BP monitoring | 0–1 |
| | Measuring of cholesterol level in patients | Cholesterol monitoring | 0–1 |
| | Providing health education | Health education | 0–1 |
| | Involvement in other health activities | Other health activities | 0–1 |
| | Special sessions or clinics for patients | Special session | 0–1 |
| Continuity | Referral decision making | Referral decision | 0–1 |
| Definition: Person-centred care where long-standing relationship between patients and healthcare providers are developed, covering substantial periods of patient's life. | Factors considered when making referral decision | Referral factors | 1–3 |
| | Past medical records received for new patients | Records continuity | 1–3 |
| | Important information included in medical records | Medical info | 0–1 |
| | Regularity of medical records keeping | Record keeping | 0–1 |
| | Discharge information received from hospital | Hosp feedback | 1–5 |
| Coordination | Practicing in a group practice | Group practice | 0–1 |
| Definition: Ensure appropriate and timely referral across all levels of care and act as care managers to their patients. | Skill mix: disciplines working in practice | Skill mix | 0–1 |
| | Involvement in disease management programme | Disease management | 0–1 |
| | Meeting face-to-face with other professionals | Meeting professions | 1–3 |
| | Seeking advice from medical specialists | Advice seeking | 1–3 |
| | Availability of nurse in practice | Nurse practice | 0–1 |
| | The use of referral letter for specialist referral | Referral letter | 1–4 |
| | Feedback from specialist for referred patients | Specialist feedback | 1–4 |

Source:

[†] World Health Organization. Draft charter for general practice/family medicine in Europe: report on a WHO meeting, Copenhagen, Denmark 6–7 February 1998

[‡] Practitioners' Questionnaire, QUALICOPC, a multi-country study evaluating quality, costs and equity in primary care.

## Comprehensiveness

From Fig 2, the public primary care clinics scored significantly higher in overall comprehensiveness of care compared to the private sector, with $\beta_{public} = 0.116$, $p<0.001$. The public sector was better in seven out of eleven indicators. However, the private sector had better access to X-ray machine, and was rated higher in providing health education to their patients compared to public sector. No significant differences were observed in cholesterol monitoring and conducting minor medical procedures.

**Table 2. Demographics details of the practitioners participated in QUALICOPC by sector.**

| Demographics | Public, N = 221 | | Private, N = 239 | | p-value |
|---|---|---|---|---|---|
| Age, Median (IQR) | 29 | (28–31) | 50 | (43–60) | < 0.0001† |
| Sex, n (%) | | | | | < 0.0001‡ |
| Men | 85 | (38.5) | 157 | (65.7) | |
| Women | 136 | (61.5) | 82 | (34.3) | |
| Practice Location, n (%) | | | | | < 0.0001‡ |
| City | 23 | (10.4) | 77 | (32.2) | |
| Suburb | 24 | (10.9) | 69 | (28.9) | |
| Town | 73 | (33.0) | 55 | (23.0) | |
| Mixed | 50 | (22.6) | 32 | (13.4) | |
| Rural | 51 | (23.1) | 6 | (2.5) | |

† Kruskal-Wallis test

‡ Chi-squared test

## Continuity

The private sector was rated significantly higher in continuity ($\beta_{public}$ = -0.028, p = 0.003) than the public sector as a whole, albeit in a small magnitude (Fig 3). The private sector achieved higher score in the indicators in related to referral decision (Referral factors and Referral decision) while the public practitioners were better at maintaining informational continuity (Record keeping, Medical info, and Records continuity).

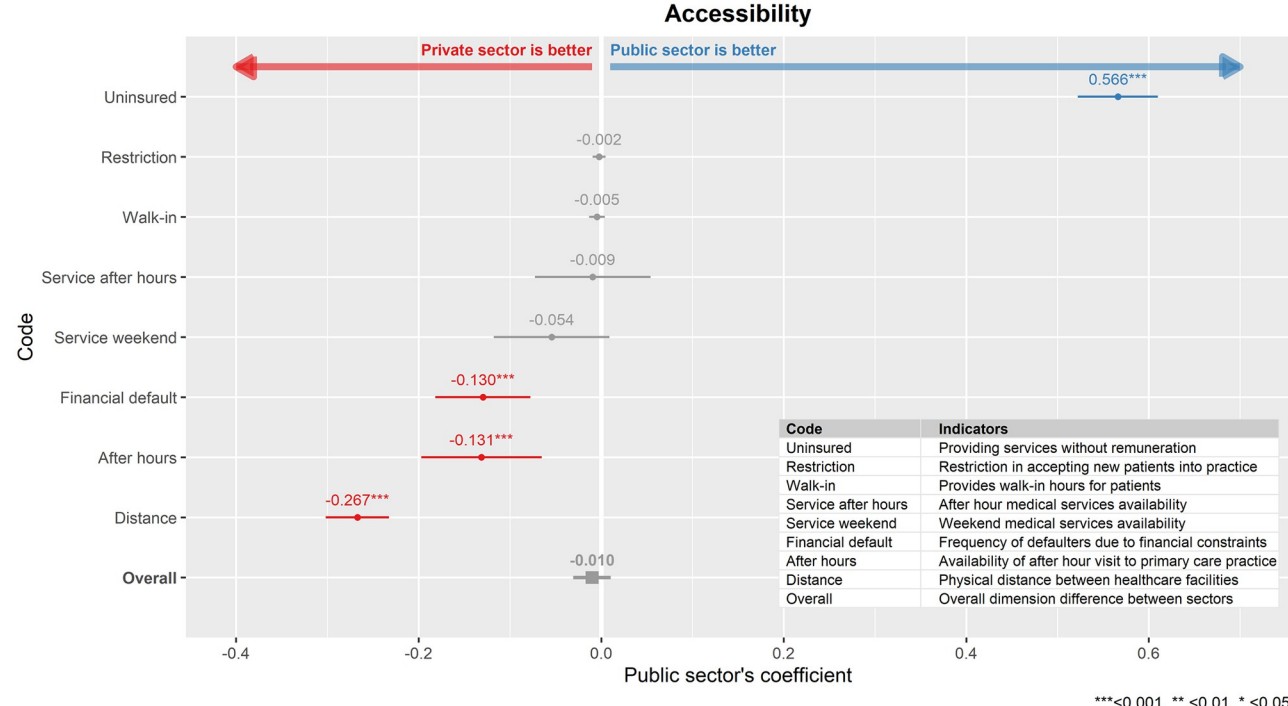

**Fig 1. Comparison between public and private primary care clinics in the dimension of accessibility.** The public sector's regression coefficient ($\beta_{public}$) for each indicator model as well as overall dimension model was shown here together, with its 95% confidence interval.

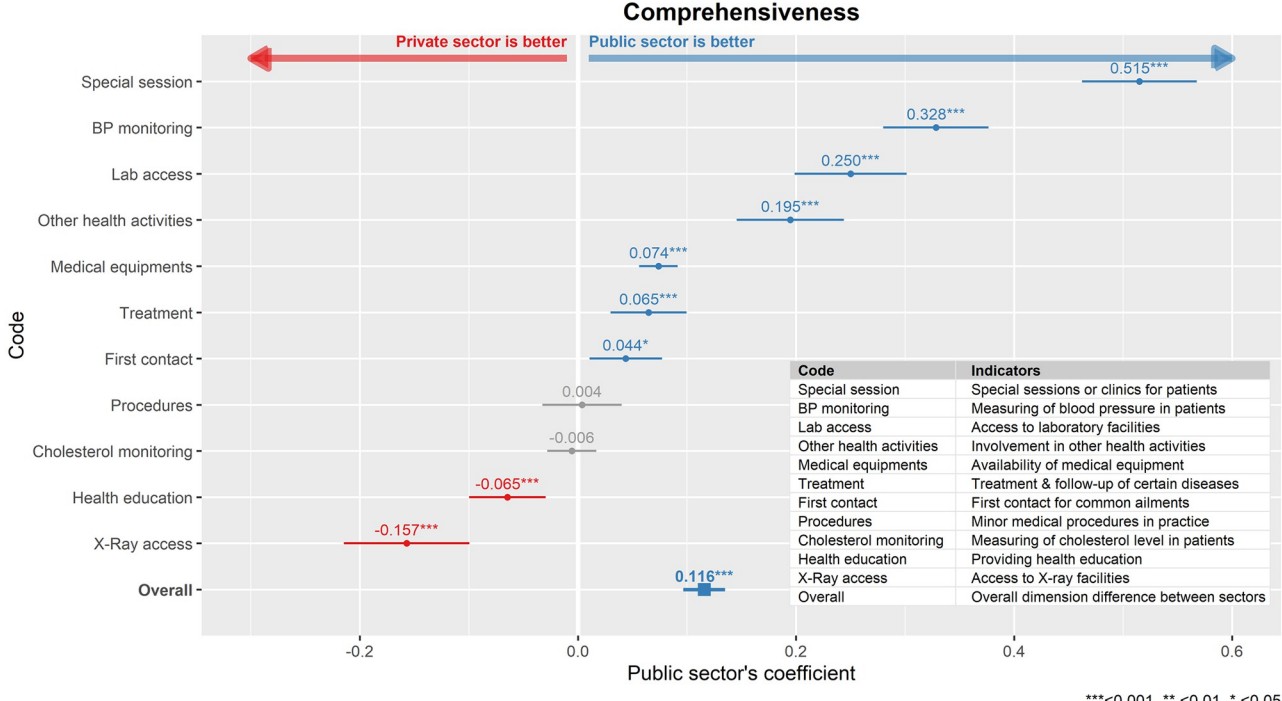

**Fig 2. Comparison between public and private primary care clinics in the dimension of comprehensiveness of care.** The public sector's regression coefficient ($\beta_{public}$) for each indicator model as well as overall dimension model was shown here together, with its 95% confidence interval.

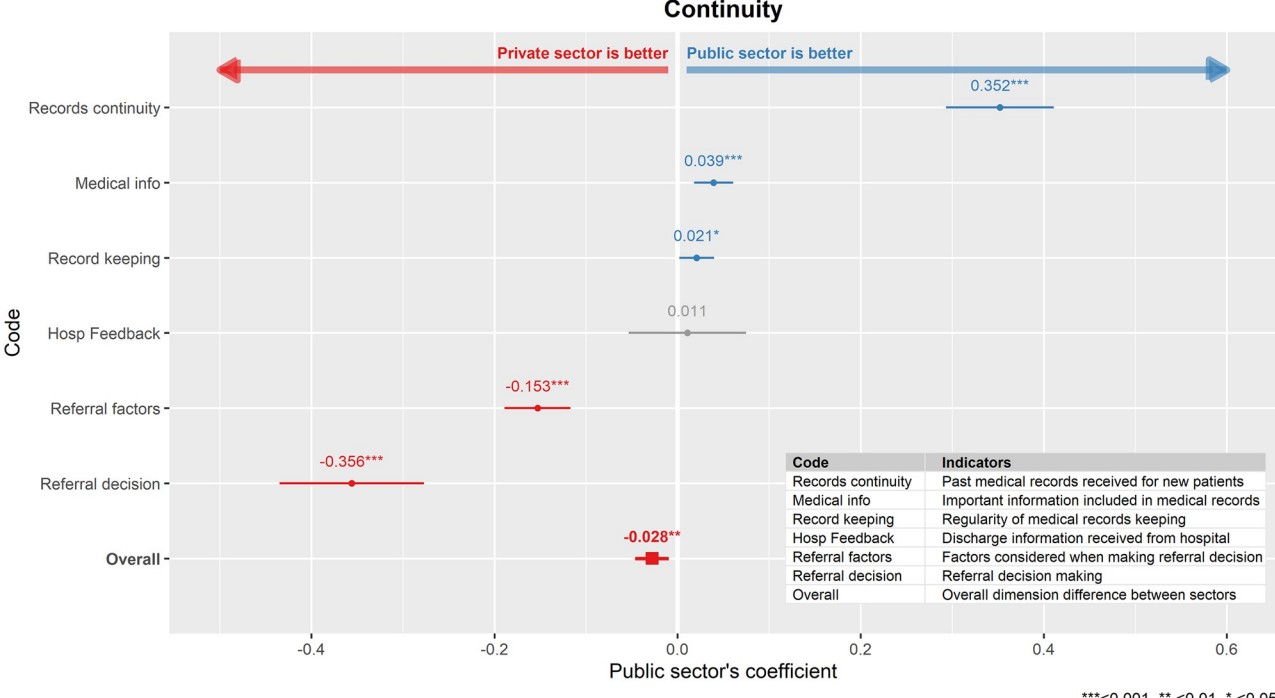

**Fig 3. Comparison between public and private primary care clinics in the dimension of continuity of care.** The public sector's regression coefficient ($\beta_{public}$) for each indicator model as well as overall dimension model was shown here together, with its 95% confidence interval.

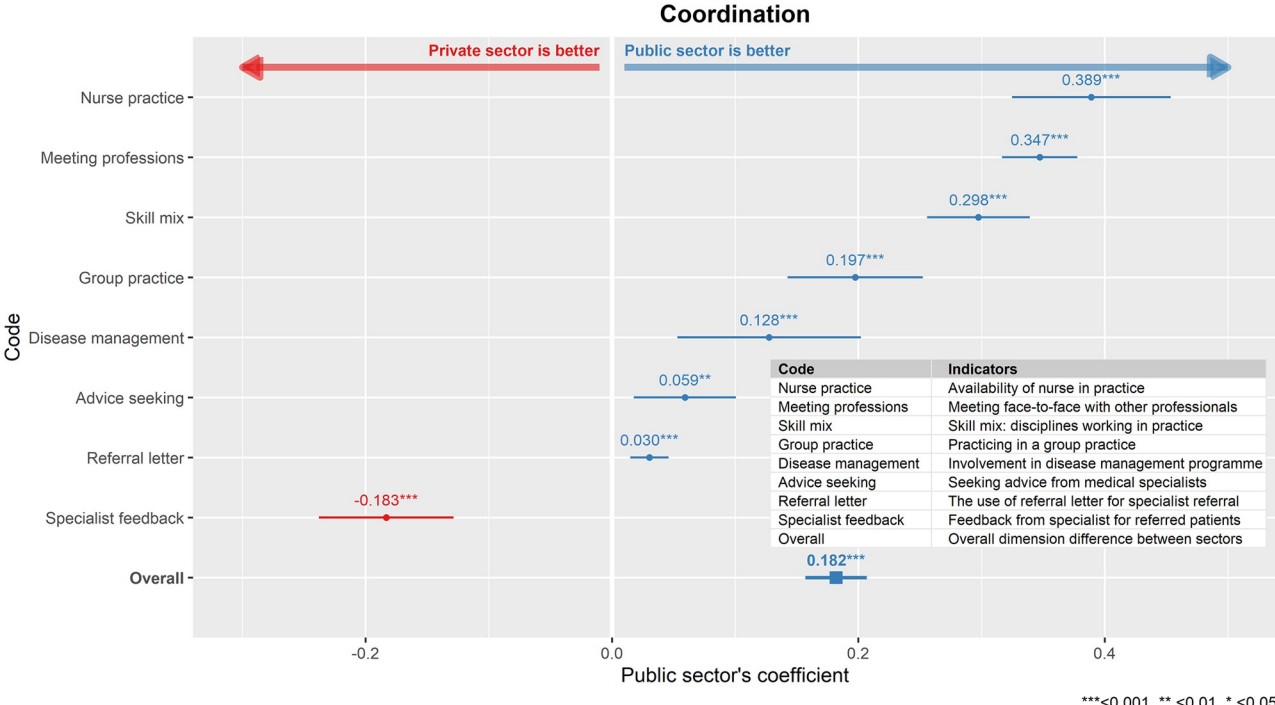

**Fig 4. Comparison between public and private primary care clinics in the dimension of coordination.** The public sector's regression coefficient ($\beta_{public}$) for each indicator model as well as overall dimension model was shown here together, with its 95% confidence interval.

### Coordination

Overall, the public sector was rate higher compared to the private sector in the dimension of coordination. The coefficient for this dimension, $\beta_{public} = 0.182$, p<0.001, was the highest among the four dimensions examined (Fig 4). Public sector was rated higher compared to private sector in seven out of eight indicators for dimension, which were related to multidisciplinary group practices. Nevertheless, receiving feedbacks from specialists was the only indicator in this dimension was rated higher in the private sector compare to public sector.

### Sensitivity analysis

The addition of sex, age and practice location did not make significant changes to the overall dimensional performances, except for continuity of care, where the overall difference between the two sectors remained the same but it had become insignificant. Most of the direction of the indicators remained consistent with the main analysis, other than Advice seeking in coordination of care, which now was in favour of private sector, but this difference was not significant. As the interest of the study lies in the overall differences between the sectors, and not the effects of practitioners' and practices' characteristics, only the main analysis results were presented and focused.

### Discussion

To the best of our knowledge, this study will be among the few in Asia to examine the differences in primary care service delivery between the public and private sectors using an international standardised questionnaire. The difference of performances in process dimensions studied were accessibility, comprehensiveness, continuity and coordination. Public sector was

rated higher in comprehensiveness and coordination, whereas, private sector was rated higher in continuity. Nevertheless, we need to examine the differences in the rating of each indicators within each of the dimensions in order to portray a more accurate picture.

## Accessibility

As a whole, there was no significant difference in the performance of accessibility domain between public and private sector primary care clinics. However, significant differences were observed when we examined the individual indicators that measured accessibility.

Firstly, the public sector practitioners were more likely to accept uninsured patients and patients with financial difficulties, thus the public sector encountered more appointment defaulters. The high appointment defaulters in the public sector could be due to personal barriers such as non-appreciative attitude towards free healthcare services, difficulty taking leave from work and transportation problems which includes cost and distance to clinics among the low socio-economic group [25]. Another possible reason is their dissatisfaction related to aspects of service quality such as clinic appointment structure and schedules, choice of provider offered, no regular doctor whom they can consult or long waiting times at the clinics [26]. These issues especially personal barriers and the long waiting times imposed high opportunity costs, may affect or discourage them from seeking care when they do not face an acute need. Secondly, the private primary care clinics were more likely to operate out of usual working hours compared to public clinics. In a previous survey, it was shown that over 90% of the private clinics in Malaysia were operational at least 6 days a week and the median working hours of the private practitioners were longer than their public counterpart [11]. Nevertheless, patients still can access out of hours public primary care services but only through specific public primary care clinics with extended hours, on-call services for emergency care in the clinics, or through the emergency services of the public hospital. Our findings were contrary to the results found in Malta where patients had found that it was harder to access out-of-hours care in the private sector [27]. Finally, as most of the private clinics were located in urban settings, it was not surprising to see that they were located closer to other healthcare facilities, thus providing many opportunities and alternatives for patients to access primary care facilities.

## Comprehensiveness

The public sector in Malaysia was significantly stronger in comprehensiveness compared to the private sectors. This finding is interesting, as it was the opposite of what was reported in Malta [27] and Hong Kong [28]. In Malta, it was reported that the patients experienced more comprehensive care in the private sector while for Hong Kong, availability and utilisation of comprehensive services were found to be better in the private sector [27, 28].

In addition, the public clinics in Malaysia were more likely to offer disease specific special sessions and other health services such as antenatal care, immunisation paediatric surveillance and palliative care. They also had better access to laboratory facilities compared to the private practitioners. This is most likely due to the difference in the structure of public and private clinics. The public clinics in Malaysia mainly exists as group practices with good skill-mix of staff and facilities [11, 12], hence this allows public clinics to offer more health activities. With this shared practices and good skill-mix, in-house laboratory access is also more readily available in the public clinics. The public practitioners were not only more likely to be the first healthcare contact but also contacted for treatment and follow-up in primary care related health problems compared to private practitioners. This is consistent with past study, which showed the primary focus of private sector was on curative care [10]. Thus, there was no

difference seen in the frequency of minor procedures, which is an indicator of curative care, between both sectors in this study. The opposite was noted in Hong Kong, where the private clinics provided more utilisation and availability for first contact care [28]. Interestingly, the private practitioners in Malaysia were more involved in giving health education compared to the public practitioners. This is possibly due to the lower patient load in the private clinics, hence allowing the practitioners to have a slightly longer consultation time [29] and include some form of health education. Another explanation is the public clinics have a multi-disciplinary team, hence some of these educational activities were handled by the pharmacists or dieticians [12]. A previous survey had shown that less than half of the primary care clinics in Malaysia had its own X-ray facilities [30], hence, the reason for the higher score in the private sector possibly boils down to its urban location where access to X-ray was more readily available.

## Continuity

Overall, the private sector performed marginally better in continuity of care than the public sector. The private sector has a much better performance in accommodating patient's needs and preferences such as shared referral decision making and consideration of patients related factors during referral. This could be attributed to the different governance in the two sectors. The private clinics operate on a fee-for service basis, are solo or small group practices, and are more inclined to have a better continuity of care with a more personalized services [31, 32]. The older median age of the private primary care doctors also probably meant a longer practice experience, which we think can foster a closer relationship between patients and doctors through regular visits. These have resulted in better interpersonal communication where the general practitioners are able to elicit and understand patient concerns and engage in shared decision making [33]. Furthermore, the private sector does not have administrative constraints in where the referral can be made. On the other hand, the implementation of shared decision-making in the public sector has been hampered by multiple challenges such as administrative constraints, different patient-doctor relationship dynamics, and lack of resources [34–36].

We found that public clinics were better at maintaining informational continuity which includes receiving past medical records, information completeness in medical records, and regularity of medical records keeping. The disparity could mainly lie with the public primary care services that are administered by a single ministry i.e., the MOH. This will facilitate information sharing between the public health facilities due to standardized information format and fewer cross-organisation barriers. Furthermore, the existence of clear guidelines on inter-facility referrals and patients transfer [37] in the public healthcare system will ensure adequate information flow when patients get transferred. In contrast, the private primary care services are delivered by numerous, individually owned, for profit clinics which might hinder information sharing either horizontally or vertically.

## Coordination

The public sector was stronger in coordination of care compared to private sector, specifically in terms of group practice, skill-mix of professions in the practice, intra- and inter-professions interactions, the use of referral letter and availability of disease management programs when compared to the private sector. In the area of professions interaction, one reason for this difference may lie in the fact of public practices mainly exist as shared or group practices. This leads to more opportunity to have frequent interactions with other primary care providers, thereby effectively facilitating coordination of care within primary care and between primary and

other levels of care. The skill-mix composition differed significantly between public and private clinics. Public clinics were staffed by a wider range of healthcare providers compared to private clinics [12]. The finding that the public sector has higher usage of referral letter could reflect the gatekeeping role of public primary care services in Malaysia [35]. While in private sector, primary care physician referral is not compulsory and patients have direct access to most specialists at secondary or tertiary private care facilities.

The higher availability of disease management programs in the public sector could be due to bulk of chronic disease management taking place in public clinics [11]. It has been reported that a comprehensive and well-coordinated care for patients with chronic conditions has been developed through the assignment of support staff and use of disease management programs [38, 39]. Hence, the solo practice structure in the private sector may limit their engagement in disease management programs for the chronically ill. The difference in receipt of feedback communication from specialists between the two sectors could be due to the structure of the services. The private healthcare sector largely operated in a fee for service funding system. As private primary care is one of the feeders to higher level private healthcare, a feedback could facilitate inter-level communication and increase collaboration and volume of consultation within the private sector.

## Implications

This study has shown that public and private primary care services have their strengths and weaknesses respectively. In short term, given the current structure of public and private divide, collaborations are best tapped into the strength of each sector. More illness and disease management that require complex program coordination to be emphasised for public sector, while curative and acute illness management to be promoted in private sector. The strength of private sector is availability of time for health education, which can be promoted as legitimate service that can be paid, or reimbursed.

Nevertheless, in the long run, few measures need to be taken to improve the overall performance of primary care service delivery. Our previous study had shown that as compared to countries with strong primary care services, Malaysia was still lagging behind in terms of comprehensiveness, continuity and coordination [15]. Specialist feedback need to be strengthened to promote seamless care, and referral decision could be more patient centred with reduced administrative constraints for public sector. While in private sector, the primary focus should be on incentivising the private providers to improve the structure, in order to provide more comprehensive care and to increase inter- and intra-professional cooperation.

## Limitation

This study has several limitations. Firstly, this is a comparison between the two parallel primary care systems in Malaysia. The results shown were relative differences and did not directly indicate the overall quality of the healthcare system in Malaysia. Secondly, as with any other self-reported survey, there might be bias due to the practitioners offering socially acceptable answers. The low response rate in the private sector might introduce selection bias although it had been taken into account in the sampling strategy by inflating the sample size of clinics. As a cross-sectional study, we are also unable to determine any causality from the findings. Finally, the findings reported by the practitioners might differ from what the patients experience when using the primary care services. A further investigation into the patients' experience when utilising primary care services can help to provide another perspective in assessing primary care service delivery performances.

## Conclusion

In conclusion, the public primary care service delivers better care in terms of comprehensiveness and coordination, while the private primary care provided better continuity of care. The findings from this study provide a good indication as to which area of improvement can be done by each sector respectively. The next step forward is looking it from the patient perspective toward utilising primary care delivery services. In the face of challenges such as ageing population, rising non-communicable disease burden, increasing health expenditure coupled with limited resources, strengthening the primary care service delivery has been shown to be one of the important steps in overcoming these challenges. By identifying the strengths and weaknesses of each sector, the government can devise a strategy to have both sectors complementing each other. This can be an integral part of the plan in improving primary care service delivery in a mixed health system setting.

## Supporting information

**S1 File.**
(PDF)

## Acknowledgments

We would like to thank the Director General of Health Malaysia for the approval to publish this manuscript. We would also like to acknowledge the contribution and support of Willemijn Schafer, and Wienke Boerma, from Netherlands Institute for Health Services Research (NIVEL) in the planning and conduct of QUALICOPC Malaysia study. We also wish to express gratitude to all primary care clinics' providers and participants who had participated in and contributed their time and information towards the study and the QUALICOPC team members for their various contributions to this project.

## Ethical approval

This study was approved by the Medical Research and Ethics Committee (MREC), Ministry of Health Malaysia as part of the Malaysia Health System Research (MHSR) with approval number NMRR-15-607-25769. All participants gave informed consent to be interviewed.

## Author Contributions

**Conceptualization:** Su Miin Ong, Ming Tsuey Lim, Seng Fah Tong, M. N. Kamaliah, Peter Groenewegen, Sheamini Sivasampu.

**Formal analysis:** Su Miin Ong, Ming Tsuey Lim, Seng Fah Tong, M. N. Kamaliah, Peter Groenewegen.

**Funding acquisition:** Sheamini Sivasampu.

**Methodology:** Su Miin Ong, Ming Tsuey Lim, Seng Fah Tong, M. N. Kamaliah, Peter Groenewegen, Sheamini Sivasampu.

**Project administration:** Sheamini Sivasampu.

**Supervision:** Sheamini Sivasampu.

**Visualization:** Su Miin Ong.

**Writing – original draft:** Su Miin Ong, Ming Tsuey Lim.

**Writing – review & editing:** Su Miin Ong, Ming Tsuey Lim, Seng Fah Tong, M. N. Kamaliah, Peter Groenewegen, Sheamini Sivasampu.

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
