## [Decision Letter · Decision Letter 0]

20 Jul 2022

PONE-D-21-17162Comparative performance of public and private primary care service delivery in Malaysia: an analysis of findings from QUALICOPCPLOS ONE

Dear Dr. Ong,

Thank you for submitting your manuscript to PLOS ONE. After careful consideration, we feel that it has merit but does not fully meet PLOS ONE’s publication criteria as it currently stands. Therefore, we invite you to submit a revised version of the manuscript that addresses the points raised during the review process.

The manuscript has been evaluated by two reviewers, and their comments are available below.

The reviewers have raised a number of concerns that need attention. They request modifications to the introduction and discussion to ensure that relevant literature is discussed and the aims of the study are clear. The reviewers also request additional methodological information, in particular they request information regarding the validation of the questionnaire tool. 

Could you please revise the manuscript to carefully address the concerns raised?

We look forward to receiving your revised manuscript.

Kind regards,

Jamie Royle

Staff Editor

PLOS ONE

Journal Requirements:

Reviewers' comments:

Reviewer's Responses to Questions

**Comments to the Author**

1. Is the manuscript technically sound, and do the data support the conclusions?

Reviewer #1: No

Reviewer #2: Yes

2. Has the statistical analysis been performed appropriately and rigorously? 

Reviewer #1: No

Reviewer #2: Yes

3. Have the authors made all data underlying the findings in their manuscript fully available?

Reviewer #1: Yes

Reviewer #2: Yes

4. Is the manuscript presented in an intelligible fashion and written in standard English?

Reviewer #1: No

Reviewer #2: Yes

5. Review Comments to the Author

Reviewer #1: This is a cross-sectional study to describe the difference between the private and public primary care in Malaysia. This topic is important. My comments are as follows:

1/ the introduction and discussion can made much more precise and concise. Some of the concepts are very unclear. For example, under discussion, "Firstly, the public sector practitioners were more likely to accept uninsured patients and patients with financial difficulties, thus the public sector encountered more financial defaulters." If they accept uninsured patients, they should have LESS financial defaulters? There are similar observations in the introduction and discussion

2/ I do not think the methodology is suitable for the research question. If comprehensiveness or continuity of the care are to be graded, these should be rated by the users (i.e. patients) and not by the healthcare providers. This may also be achieved by direct observation of the service by an independent raters

3/ How are the sample sizes of both private or public doctors achieved?

4/ I understand that the questionnaire is being used in several countries. However, is this validated? Similarly, is the way to combine the scores from each questions validated?

Reviewer #2: The objective of this study was to determine the difference in primary care services delivery dimensions between public and private clinics. Data from the Malaysian International Quality and Costs of Primary Care (QUALICOPC) study were used.

This is a relevant and well conducted study. I have only a few minor comments and suggestions.

1. The information of the data collection (sampling and questionnaire) is very concise. The authors refer to other publications for more details, however, the reader of this manuscript should be able to judge the result without having to read additional articles. Some examples were more information is needed:

- sampling method: “222 public primary care practitioners were sampled” and “239 practitioners responded to the survey out of the 510 clinics sampled”. Was the sampling frame for the public primary care a list of practitioners and for the private primary care a list of clinics?

- Page 21: the authors wrote “The low response rate in the private sector might introduce selection bias although it had been taken into account in the sampling strategy.”, however, no information is available about the sampling strategy.

- No information is available of the other variables in the questionnaire.

2. Page 16: the authors wrote: “this study is the first in Asia to examine the differences in primary care service delivery between the public and private sectors using an international standardised questionnaire.”. However this research question can also found in this study: Nguyễn Thị, Hòa, et al. “Patient Experiences of Primary Care Quality amongst Different Types of Health Care Facilities in Central Vietnam.” BMC HEALTH SERVICES RESEARCH, vol. 19, 2019, doi:10.1186/s12913-019-4089-y.

6. PLOS authors have the option to publish the peer review history of their article (what does this mean?). If published, this will include your full peer review and any attached files.

Reviewer #1: No

Reviewer #2: **Yes: **Wim Peersman

---

## [Author Response · Author response to Decision Letter 0]

6 Sep 2022

Authors’ response to reviewer#1

1) The introduction and discussion can made much more precise and concise. 

Thank you for the suggestions. 

We have rephrased and combined paragraph 2 and 4 which becomes paragraph 3 in the introduction section to make it more concise. The previous paragraph 3 was moved to paragraph 2 (page 4, line 77-97)

It now reads:

Introduction, page 5, line 98-106 

“In countries with mixed health systems, the private sector is often perceived to offer access to greater service capacity and responsiveness, managerial expertise, technology and innovation as well as investment and funding [6]. Hence, it is imperative to understand how the quality and performance in the private sector compares with that in the public sector. It is important to have an optimal balance of roles and facilities in the provision of health services between public and private sectors. By identifying the strengths and weaknesses of the service delivery process of the two sectors, the gaps between the sectors can be bridged in anticipation of possible future public-private partnership and lead towards a better holistic health system for all patients.”

- Some of the concepts are very unclear. For example, under discussion, "Firstly, the public sector practitioners were more likely to accept uninsured patients and patients with financial difficulties, thus the public sector encountered more financial defaulters." If they accept uninsured patients, they should have LESS financial defaulters? There are similar observations in the introduction and discussion

We agree with the reviewer that some of the concepts are confusing. We have corrected it in the following statements.

It now reads:

Discussion, page 16, line 268-278

“Firstly, the public sector practitioners were more likely to accept uninsured patients and patients with financial difficulties, thus the public sector encountered more appointment defaulters. The high appointment defaulters in the public sector could be due to personal barriers such as non-appreciative attitude towards free healthcare services, difficulty taking leave from work and transportation problems which includes cost and distance to clinics among the low socio-economic group [25]. Another possible reason is their dissatisfaction related to aspects of service quality such as clinic appointment structure and schedules, choice of provider offered, no regular doctor whom they can consult or long waiting times at the clinics [26]. These issues especially personal barriers and the long waiting times imposed high opportunity costs, may affect or discourage them from seeking care when they do not face an acute need.”

2) I do not think the methodology is suitable for the research question. If comprehensiveness or continuity of the care are to be graded, these should be rated by the users (i.e., patients) and not by the healthcare providers. This may also be achieved by direct observation of the service by an independent raters

Thank you for pointing this out.

We acknowledge your comments and agree that that comprehensiveness and continuity of care can be rated by the users, which was captured by the patient experience part of the questionnaires. 

At the same time, healthcare providers are well positioned to observe these aspects of care. We also think that by surveying the providers, we can gauge the availability of services provided by the primary care service provider which patients might not be aware. We can also know if the current structure and process in the providers services are facilitating the continuity of care among their patients (ref1, ref 2, ref3). Therefore, in this study we first examine these care services from the healthcare providers’ viewpoints.

We included a statement on the utilization of care services from the perspective of healthcare providers and patients in the introduction section and conclusion section respectively.

It now reads: 

Introduction, page 6, line 114-116 

“Therefore, in this study, we aimed to use the data from QUALICOPC to answer the question, is there a difference in primary care services delivery dimensions between public and private clinics from the perspective of providers?”

Conclusion, page 22, line 403-404

“The next step forward is looking it from the patient perspective toward utilising primary care delivery services.”

References

ref1) Pavlič, D., Sever, M., Klemenc-Ketiš, Z., Švab, I., Vainieri, M., Seghieri, C., & Maksuti, A. (2018). Strength of primary care service delivery: A comparative study of European countries, Australia, New Zealand, and Canada. Primary Health Care Research & Development, 19(3), 277-287. doi:10.1017/S1463423617000792

ref2) Pavlič, D.R., Sever, M., Klemenc-Ketiš, Z. et al. Process quality indicators in family medicine: results of an international comparison. BMC Fam Pract 16, 172 (2015). https://doi.org/10.1186/s12875-015-0386-7

ref 3) Lim MT, Ong SM, Tong SF, et al. Comparison between primary care service delivery in Malaysia and other participating countries of the QUALICOPC project: a cross-sectional study. BMJ Open 2021; 11: e047126.

3) How are the sample sizes of both private or public doctors achieved?

The calculation was made based on the target sample size of 220 clinics similar to other countries which also participated in QUALICOPC survey for benchmarking. In sample size calculation, we have taken into account of the high non-response rate which was based on a response rate of a previous study among public and private clinics in Malaysia (17). We have inflated sample size for private clinics from targeted 220 to overcome the low response rate in private clinics. Furthermore, various strategies including obtaining support letters from the Deputy Director General of Health and Technical Support (DDG), state health directors as well as the Malaysian Medical Association (MMA) and the Board of Directors of large chain clinics were taken to increase the doctor’s participation in the private sector.

4) I understand that the questionnaire is being used in several countries. However, is this validated? Similarly, is the way to combine the scores from each questions validated?

For the first question, we have conducted face and content validation. The questionnaires were validated by two family medicine specialists and three researchers through several consensus round. The items/indicators in the questionnaire pertaining to the dimensions of accessibility, comprehensiveness, continuity and coordination of care were in accordance to the WHO definition of characteristics of primary care (27). We have used the same measurements in our previous study which demonstrated the differences between countries for these concepts, thus providing evidence for its content validated (ref1).

We have added these sentences on validation to the methods section to add clarity. 

It now reads:

Materials and methods: data source, page 7, line 126-130

“The questionnaires were validated by family medicine specialists and healthcare researchers for face and content validity. We have used the same measurements in our previous study which demonstrated the differences between countries for these concepts, thus providing evidence for its content validated [15].”

We have now included the questionnaires as a supplementary material.

Next, the ecometric approach in combining the scores using multilevel linear regression was shown to be a valid method in constructing contextual variables from individual responses (29). A separate study using QUALICOPC questionnaire was using a similar approach in aggregating the scores (ref1, ref2).

References

ref1) Lim MT, Ong SM, Tong SF, et al. Comparison between primary care service delivery in Malaysia and other participating countries of the QUALICOPC project: a cross-sectional study. BMJ Open 2021; 11: e047126.

ref2) Pavlič, D., Sever, M., Klemenc-Ketiš, Z., Švab, I., Vainieri, M., Seghieri, C., & Maksuti, A. (2018) Strength of primary care service delivery: A comparative study of European countries, Australia, New Zealand, and Canada. Primary Health Care Research & Development, 19(3), 277-287. doi:10.1017/S1463423617000792

Author’s response for reviewer#2

This is a relevant and well conducted study. I have only a few minor comments and suggestions.

1. The information of the data collection (sampling and questionnaire) is very concise. The authors refer to other publications for more details, however, the reader of this manuscript should be able to judge the result without having to read additional articles. Some examples were more information is needed:

sampling method: “222 public primary care practitioners were sampled” and “239 practitioners responded to the survey out of the 510 clinics sampled”. Was the sampling frame for the public primary care a list of practitioners and for the private primary care a list of clinics? 

- Page 21: the authors wrote “The low response rate in the private sector might introduce selection bias although it had been taken into account in the sampling strategy.” however, no information is available about the sampling strategy.

- No information is available of the other variables in the questionnaire.

Thank you for the suggestion.

We have added more details as suggested. 

It now reads:

Materials and Methods: data source, page 7-8, line 133—150.

“The list of primary care clinics was obtained from the Ministry of Health Malaysia through the Family Health Development Division for public clinics and the Private Medical Practice Control Division for private clinics. If a sampled clinic had more than one doctor working on that day of data collection, only one primary care doctor per clinic would be selected to participate in the study. The doctor in the public clinics was selected by simple random sampling, whereas in the private clinics, the doctor was selected by convenient sampling [16]. A total of 222 public primary care practitioners were sampled with 221 responded (response rate of 99.5%). As for private sector, 239 practitioners responded to the survey out of the 510 clinics sampled, giving a response rate of 46.9%. The calculation was made based on the target sample size of 220 clinics similar to other countries which also participated in QUALICOPC survey for benchmarking. We are also taking into account the high non-response rate of a previous local study among public and private clinics in Malaysia [11]. Hence, the sample size for private clinics was inflated from targeted 220 to overcome the low response rate in private clinics. Various strategies including obtaining support letters from the Deputy Director General of Health and Technical Support (DDG), state health directors as well as the Malaysian Medical Association (MMA) and the Board of Directors of large chain clinics were taken to increase the doctor’s participation in the private sector.”

-To overcome the low response rate in private sector, the calculate sample size for private clinics were inflated from targeted 220. The final samples of clinics completed the study was 226.

We have rephrased the sentence in the limitation section to add further clarity. 

It now reads:

Discussion: Limitation, page 22, line 392-394

“The low response rate in the private sector might introduce selection bias although it had been taken into account in the sampling strategy was overcome by inflating the sample size of clinics.”

-The questionnaires have a comprehensive list of questions that cover most of the aspects for primary care services. We have now included the questionnaires as a supplementary material.

2. Page 16: the authors wrote: “this study is the first in Asia to examine the differences in primary care service delivery between the public and private sectors using an international standardised questionnaire.”. However, this research question can also found in this study: Nguyễn Thị, Hòa, et al. “Patient Experiences of Primary Care Quality amongst Different Types of Health Care Facilities in Central Vietnam.” BMC HEALTH SERVICES RESEARCH, vol. 19, 2019, doi:10.1186/s12913-019-4089-y.

Thank you for pointing this out. We apologize for the oversight. We have amended it.

It now reads:

Discussion, page 16, line 256-258

“To the best of our knowledge, this study will be among the few in Asia to examine the differences in primary care service delivery between the public and private sectors using an international standardised questionnaire.”

---

## [Editor Report · Decision Letter 1]

10 Oct 2022

Comparative performance of public and private primary care service delivery in Malaysia: an analysis of findings from QUALICOPC

PONE-D-21-17162R1

Dear Dr. Ong,

We’re pleased to inform you that your manuscript has been judged scientifically suitable for publication and will be formally accepted for publication once it meets all outstanding technical requirements.

Kind regards,

Tarik A. Rashid, PhD

Academic Editor

PLOS ONE

---

## [Editor Report · Acceptance letter]

13 Oct 2022

PONE-D-21-17162R1 

Comparative performance of public and private primary care service delivery in Malaysia: an analysis of findings from QUALICOPC 

Dear Dr. Ong:

I'm pleased to inform you that your manuscript has been deemed suitable for publication in PLOS ONE. Congratulations! Your manuscript is now with our production department. 

Kind regards, 

on behalf of

Dr. Tarik A. Rashid 

Academic Editor

PLOS ONE